# A modular platform for one-step assembly of multi-component membrane systems by fusion of charged proteoliposomes

Robert R. Ishmukhametov[1], Aidan N. Russell[1] & Richard M. Berry[1]

An important goal in synthetic biology is the assembly of biomimetic cell-like structures, which combine multiple biological components in synthetic lipid vesicles. A key limiting assembly step is the incorporation of membrane proteins into the lipid bilayer of the vesicles. Here we present a simple method for delivery of membrane proteins into a lipid bilayer within 5 min. Fusogenic proteoliposomes, containing charged lipids and membrane proteins, fuse with oppositely charged bilayers, with no requirement for detergent or fusion-promoting proteins, and deliver large, fragile membrane protein complexes into the target bilayers. We demonstrate the feasibility of our method by assembling a minimal electron transport chain capable of adenosine triphosphate (ATP) synthesis, combining *Escherichia coli* $F_1F_o$ ATP-synthase and the primary proton pump $bo_3$-oxidase, into synthetic lipid vesicles with sizes ranging from 100 nm to $\sim$10 µm. This provides a platform for the combination of multiple sets of membrane protein complexes into cell-like artificial structures.

[1] Department of Physics, University of Oxford, Clarendon Laboratory, Parks Road, Oxford OX1 3PU, UK. Correspondence and requests for materials should be addressed to R.R.I. (email: Robert.Ishmukhametov@physics.ox.ac.uk) or to R.M.B. (email: Richard.Berry@physics.ox.ac.uk).

The vast majority of purified membrane proteins are incapable of self-insertion into lipid bilayers. Therefore, reconstituting membrane protein complexes in lipid bilayers for *in vitro* studies becomes critically limiting as the complexity of the system to be reconstituted increases. The most popular reconstitution method, addition and subsequent removal of detergent at above the critical micelle concentration[1] works well with pre-formed small unilamellar vesicles (SUV, or liposomes, 30–200 nm in diameter), but is less effective with large unilamellar vesicles (LUV, 200–1,000 nm) and is deleterious to giant unilamellar vesicles (GUV, >1,000 nm). More advanced planar bilayer systems, for example Droplet on Hydrogel Bilayer[2] or tissue-like structures[3] would not tolerate detergents. Various laborious and/or time-consuming techniques have been developed to address this challenge[4–7], but fast, easy and gentle incorporation of membrane proteins into large bilayers remains problematic.

An alternative to the use of detergents is vesicle fusion, which is less harmful to membrane proteins and the bilayer integrity and may allow delivery of proteoliposome-incorporated proteins into much larger accepting bilayers, retaining orientation of membrane proteins in the bilayer. During vesicle fusion two interacting lipid bilayers combine to form a continuous post-fusion bilayer, and their internal contents mix without release to the surrounding medium. Fusion requires the interacting bilayers to be brought into very close proximity[8,9], overcoming the electrostatic repulsion between typically negatively charged lipid heads which otherwise makes fusion impossible. If too much repulsion remains, this can lead to aggregation of vesicles followed by hemifusion: when the external lipid monolayers of the interacting vesicles unite but the inner monolayers do not, keeping the liquid contents separated.

How repulsion is overcome depends on the nature of the interacting bilayers and the lipid fusion system. *In vivo* vesicle fusion is driven by a large variety of fusion-promoting complementary membrane proteins found in viruses[10] and intracellular organelles[11–14], which must be present in both interacting membranes and which pull the bilayers toward each other in a fusion complex[15,16]. Some of these proteins[17–19] have been used for vesicle fusion *in vitro* but a limitation of this approach is that the accepting bilayer a priori must have the complementary protein in the membrane. It can be overcome by using complementary DNA oligonucleotides[20,21], which are designed to insert themselves into the lipid bilayer and drive vesicle fusion as they hybridize and pull the membranes towards each other. However *in vitro* fusion by both methods is relatively slow, requiring on the order of an hour[17,20].

Much faster vesicle fusion was demonstrated between vesicles formed of complementary charged lipids. Such vesicles have been used as miniature confined reactors for chemical reactions[22] and in vesicle fusion studies[23–25]. They fuse within milliseconds of encountering each other[26], with either hemi-fusion or full fusion as the end-state, depending on the relative content of charged lipids in the membrane[27,28] and the ionic strength[29] of the external medium. In general fusion is promoted by low ionic strength, when ions do not shield the attractive interaction between oppositely charged lipid heads. Nevertheless, despite its simplicity, complementary charged lipids have not previously been used as a method for the fast delivery of transmembrane proteins from one lipid object to another.

$F_1F_o$ Adenosine triphosphate (ATP) -synthase, which makes most cellular ATP in living organisms, is of great interest in synthetic biology as a renewable source of ATP to power metabolic reactions in bio-synthetic networks[30,31]. This sophisticated rotary molecular machine[32,33] is the link between the primary (proton motive force, PMF) and secondary (ATP) forms of biological free energy. Depending on physiological conditions it either makes ATP at the expense of PMF by using adenosine diphosphate (ADP) and inorganic phosphate ($P_i$), or generates PMF by cleaving ATP into ADP and $P_i$, using $Mg^{2+}$ as a cofactor for both reactions. *Escherichia coli* ATP-synthase is notoriously difficult to handle because it quickly loses integrity during lengthy isolation or if exposed to heat, making it a demanding test of the power of our method. By contrast, $bo_3$-oxidase[34] is a robust powerful primary proton pump found in bacterial electron transport chains, and can be used to generate a PMF across a lipid bilayer. This redox protein uses natural membrane quinols like Coenzyme $Q_{10}$ as a donor and oxygen as an acceptor of electrons. For both proteins, detailed and highly reproducible isolation and reconstitution procedures are well established[35,36].

Here we explore the possibility of using complementary charged lipids and demonstrate that they can be used for facile and rapid delivery of functional transmembrane proteins into existing lipid bilayers of various sizes, including SUV, LUV and GUV. To demonstrate the versatility of our method, we assembled a bio-mimetic functional system by reconstituting two different detergent-solubilised membrane protein complexes into charged SUV, and fusing them with oppositely charged bilayers of various sizes to gain a measurable biological function impossible for the individual components. We chose *E. coli* $F_1F_o$ ATP-synthase (Mw 520 KDa, 22 subunits) and $bo_3$-oxidase (Mw 100 KDa, four subunits) as model large multi-subunit membrane protein complexes, which can combine to form a minimal electron transport chain capable of ATP synthesis. We believe that our approach will be useful in a broad range of applications where fast reconstitution of membrane proteins is desired, and be an important tool for synthetic biology.

## Results

**Fusion of complementary charged vesicles**. We used three different lipid compositions in this study (see Methods for full names of lipids): (1) for neutral SUV/LUV/GUV ('xUV⁰') and proteoliposomes ($PL^0$), zwitterionic PC; (2) for anionic xUV ($xUV^-$) and proteoliposomes ($PL^-$), 75% PC (by weight) combined with 25% anionic lipid POPA; (3) for cationic xUV ($xUV^+$) and proteoliposomes ($PL^+$), 50% PC combined with 50% cationic lipid E-PC[37] or DOTAP[38]. These cationic lipids are used widely to form unilamellar SUV[27,39]. The lipid compositions were designed to form vesicles that fuse when mixed in the combination $2 + 3$ but no other. This allows for the construction of stable objects that fuse only when mixed. We optimized empirically the speed and yield of fusion products and functionality of membrane proteins by adjustment of lipid compositions and ionic strength of the reaction medium.

We characterized fusion of vesicles by monitoring liquid content transfer, liquid content leakage, intervesicular lipid mixing and inner monolayer lipid mixing. We demonstrated liquid content mixing as an indication of full vesicle fusion, using the calcein-cobalt method[26]. We formed anionic vesicles of various sizes containing EDTA, and cationic SUV by extrusion in the presence of the fluorophore calcein complexed with cobalt $Co^{2+}$ ions, which quenches calcein fluorescence. Vesicle fusion with content mixing upon combining these vesicles in media lacking free calcein is indicated by an increasing fluorescent signal (Fig. 1a), as EDTA preferentially chelates $Co^{2+}$ leaving calcein free to fluoresce inside the fusion product. Figure 1b, red shows vesicle fusion versus time following mixing of cationic and anionic 200 nm SUV in a dilute buffer (1 mM MOPS pH 7.4),

indicating rapid vesicle fusion. Fusion (%) was calculated from calcein fluorescence intensity, and calibrated by adding detergent to release and thus mix all vesicle contents (Methods, Supplementary Fig. 1). ~20% vesicle fusion is similar to published values for similar content of charged lipids[26,27].

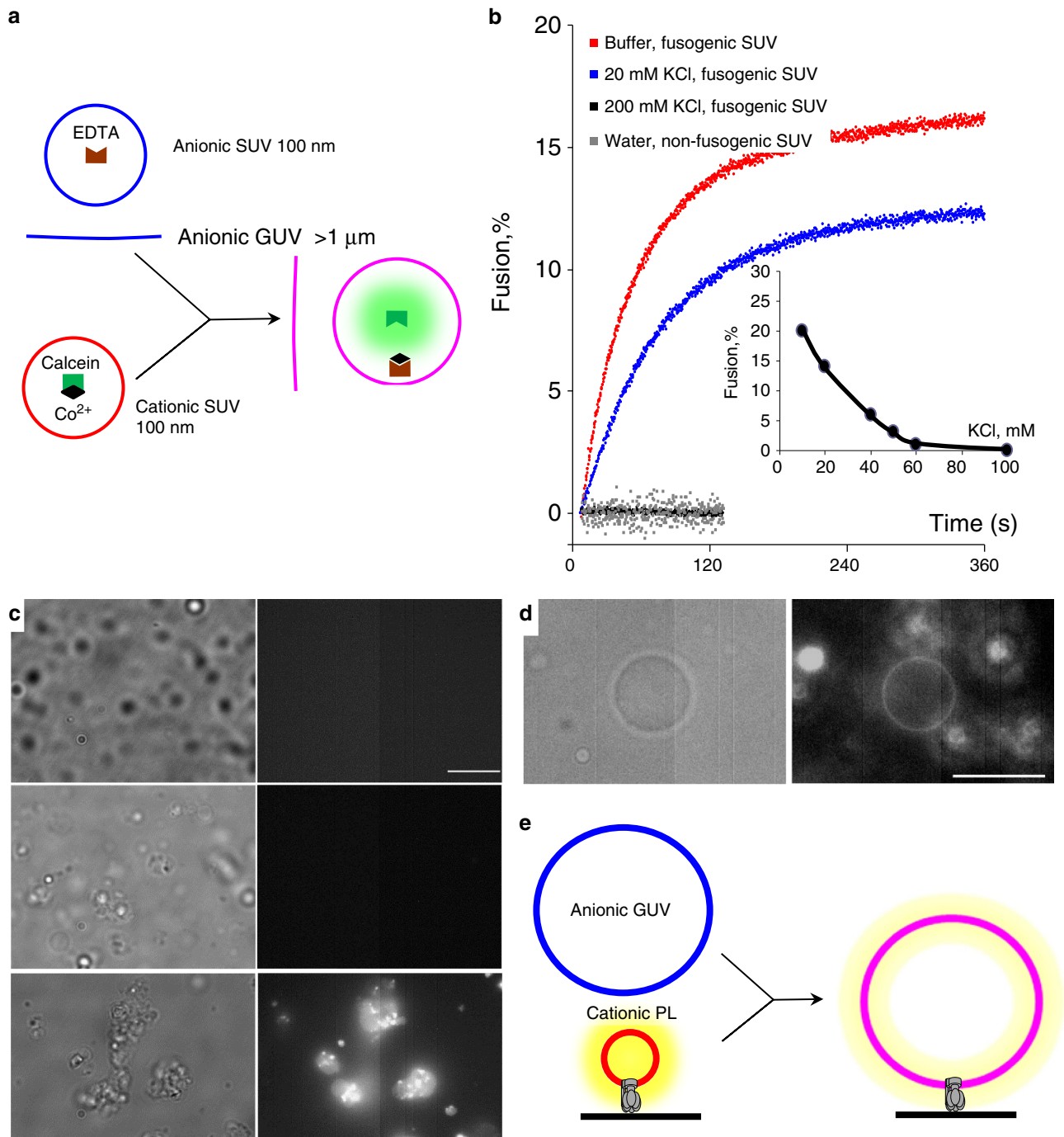

**Figure 1 | Fusion of lipid vesicles studied with cobalt-calcein liquid content transfer assay. (a)** Schematic representation of the assay. Fusion of non-fluorescent cobalt-calcein loaded positively charged small unilamellar vesicles (SUV+) (red) with EDTA-loaded negatively charged vesicles of various sizes (blue) monitored by calcein fluorescence in the fusion product (pink) upon liquid content mixing. **(b)** Calcein fluorescence versus time upon mixing suspensions of oppositely charged SUV in buffer (1 mM MOPS, pH 7.4) (red) or buffer with 20 (blue) or 200 (black) mM KCl. Increasing fluorescence indicates vesicle fusion. Replacing cationic with neutral vesicles gives no fusion (grey). The inset: % of fusion as a function of KCl concentration. **(c)** Fusion between SUV+ and GUV−, in bright-field (left) and epi-fluorescence (right) microscopy. Post-fusion vesicles containing free calcein form fluorescent clumps (bottom), which are absent in controls lacking GUV− (top) or SUV+ (middle). **(d)** A surface-immobilized anionic GUV, following fusion with one or more cationic proteoliposomes (PL+) anchored to the surface by ATP-synthase in 20 mM KCl, 1 mM MOPS pH 7.4, in bright-field (left) and fluorescence (right). Fluorescence is due to fluorescent cholesterol originally included in the PL+ membranes. Unfused PL+ are visible out of focus in the background. Fluorescence of the GUV membrane indicates fusion with the PL+. Scale bars are 10 μm in **c,d**. **(e)** A schematic of the experiment illustrated in **d**.

A control replacing anionic with neutral SUV shows no interaction between vesicles (black trace on Fig. 1b). Adding 20 mM KCl to this buffer screens the electrostatic attraction between oppositely charged vesicles, slowing the rate of fusion (Fig. 1b, blue). Further KCl addition abolishes fusion (grey) with a critical concentration being around 60 mM (Fig. 1b, inset). Addition of salt to 50, 100 and 150 mM KCl (quantities sufficient to inhibit fusion when present before addition of the vesicles, Fig. 1b, inset) 30 s after complementary vesicles were mixed in low-salt buffer did not stop fusion (Supplementary Fig. 2, red arrows).

Liquid content leakage was ~4% during complementary vesicle fusion in low salt, and negligible in high salt or with $SUV^+$ replaced by non-fusogenic $SUV^0$ (Methods, Supplementary Fig. 3).

Although complementary vesicle fusion is abolished by > ~60 mM KCl, $SUV^+$ and $SUV^-$ mixed in high salt still aggregate (as indicated by light scattering at 272 nm (ref. 40) (Supplementary Fig. 4, red) and show pronounced intervesicular lipid mixing (Methods, Supplementary Fig. 5a, blue). $SUV^0$ and $SUV^-$ show neither aggregation nor lipid mixing (Supplementary Figs 4 and 5a, black traces). The extent of inner lipid monolayer mixing during fusion in low salt is similar to that of content mixing (Methods, Supplementary Fig. 5b), consistent with ~20% full fusion. Intervesicular lipid mixing at ~50% in high or low salt indicates a further population of vesicles that can exchange lipids but not contents. Surprisingly, some inner lipid monolayer mixing remains in high salt (Supplementary Fig. 5b) despite the absence of full fusion.

These observations indicate that complementary charged vesicles mixed in low salt rapidly form a complex that is committed to subsequent full fusion regardless of later salt addition, while vesicles mixed in higher salt form an aggregated or hemi-fused complex that does not proceed to full fusion, but allows a high extent of lipid mixing, including some mixing of inner lipid leaflets. Full fusion is the rate-limiting step, taking ~1 min to reach half maximum and 7–10 min to complete (Fig. 1b), while hemifusion and vesicle rupture happen practically instantaneously, taking 1–3 s to reach half-maximum (red traces on Supplementary Figs 5 and 3).

Since non-polar osmolytes like glucose and sucrose are often used for GUV formations[5–7,] we tested their influence on fusion. We found that 40–100 mM glucose enhanced fusion (Supplementary Fig. 6), and combination of 80 mM glucose with 20 mM KCl made the fusion yield similar to that in low-salt buffer alone. In contrast, sucrose inhibited fusion at levels similar to half the molar concentration of KCl. We do not know why sucrose and glucose have opposite effects.

Figure 1c–e shows fusion between $SUV^+$ and $GUV^-$, visualised directly with epi-fluorescence microscopy. Post-fusion vesicles containing free calcein form fluorescent clumps (Fig. 1c, bottom) after mixing suspensions of cationic SUV with anionic GUV. Controls omitting $GUV^-$ or $SUV^+$ (Fig. 1c; top, middle, respectively) show no fluorescence, indicating no fusion. Formation of large clumps can be avoided by running fusion on a surface, as previously shown for SUV fusion[22]. Figure 1d,e shows a combination of this approach with the delivery of a large membrane-protein complex by vesicle fusion. $PL^+$ were formed by reconstitution of $F_1F_o$ ATP-synthase into $SUV^+$ formed in the presence of 0.5% BODIPY-FL fluorescent cholesterol, to label the $PL^+$ membrane (Methods). These were anchored via 6-histidine tags on the β-subunits of ATP-synthase to a glass surface modified with NTA-functionalized poly(L-lysine)-g-poly(ethylene glycol)[41] in a flow-cell. Free $PL^+$ were washed away by flow and then $GUV^-$ were gently flowed in and left for 10 min to fuse with the anchored $PL^+$ in 20 mM KCl, before washing away free

GUV. After fusion the fluorescent cholesterol is spread evenly in the post-fusion membranes thus making visible the otherwise non-fluorescent GUV. Figure 1d shows a post-fusion GUV in bright-field (left) and fluorescence (right). Out-of-focus surface immobilized $PL^+$ are also visible in the background. Figure 1e shows a schematic representation of this experiment.

**Effect of bilayer charge on ATP-synthase.** Although cationic fatty acid amines and alcohols exist in some eukaryotic membranes[42,43], eubacterial cationic phospholipids have not been discovered so far, and the functionality of ATP-synthase and $bo_3$-oxidase in cationic bilayers is not described in the literature. By contrast, anionic lipids constitute up to 30% of the *E. coli* cytoplasmic membrane[44], where these enzymes naturally occur, and combinations of PC and phosphatidic acid are widely used as standard components of lipid mixtures for reconstitution of ATP-synthase from various species[45,46]. Therefore it is reasonable to expect that the enzymes would perform better in native-like anionic and/or neutral lipid environments than in cationic lipids. We tested this by reconstituting the proteins into cationic, anionic and neutral proteoliposomes and assessing their functionality.

After reconstitution ~95% of ATP-synthase is oriented so as to pump protons into proteoliposomes when driven by ATP hydrolysis, that is, with $F_1$ complex facing outwards[47]. To confirm this we quantified ATP hydrolase activity of freshly made proteoliposomes with and without vesicle disruption by the non-denaturing detergent 0.5% sodium cholate (Methods). We saw no increase with detergent, indicating the absence of an inward facing fraction of $F_1$ that would have been released by vesicle disruption.

Proton pumping into PL can be easily assessed by monitoring quenching of a pH sensitive fluorescent probe 9-Amino-6-Chloro-2-Methoxyacridine (ACMA), a standard test used in such studies. ACMA quenching can be reversed by dissipating the pH gradient with uncouplers like nigericin. We reconstituted functional ATP-synthase with $SUV^+$ formed with 50% DOTAP or E-PC. ACMA quenching driven by ATP hydrolysis in these $PL^+$ (Supplementary Fig. 7a) was retained for at least 3 days if stored at room temperature or on ice. Yield of ATP-synthase reconstitution was similar for PC and E-PC proteoliposomes as judged by similar total ATP hydrolase activity in presence of 0.4% LDAO (N,N-dimethyldodecylamine N-oxide), a well-known specific activator of ATP hydrolysis (Supplementary Fig. 7b). In buffers containing 100 mM KCl, DOTAP $PL^+$ showed less ACMA quenching than those formed with E-PC, which in turn showed less quenching than $PL^0$ formed with pure PC. We note that DOTAP is a synthetic non-triglyceride lipid, while E-PC closely resembles natural PC in its structure and characteristics[37], which might be more compatible with ATP-synthase function.

Since vesicle fusion requires low ionic strength (Fig. 1), we checked the dependence of ACMA quenching by ATP-synthase on concentration of KCl. This was negligible for $PL^0$ (Supplementary Fig. 8a) in the range 1–20 mM KCl. In $PL^+$ ACMA quenching declined almost to zero with decreasing KCl concentration between 10 and 1 mM (Supplementary Fig. 8b). This effect was reversible; addition of 50 mM KCl (blue arrow) restored ACMA quenching.

Similar dependence on lipid charge and ionic strength was found for proton pumping by $bo_3$-oxidase (Supplementary Fig. 8c).

**Delivery of ATP-synthase into bilayers by vesicle fusion.** The experiment illustrated in Fig. 1d indicates that we can deliver ATP-synthase from $PL^+$ into anionic vesicles by vesicle fusion. However, that experiment does not demonstrate functionality of ATP-synthase in the post-fusion GUV. Figure 2a (pink) shows

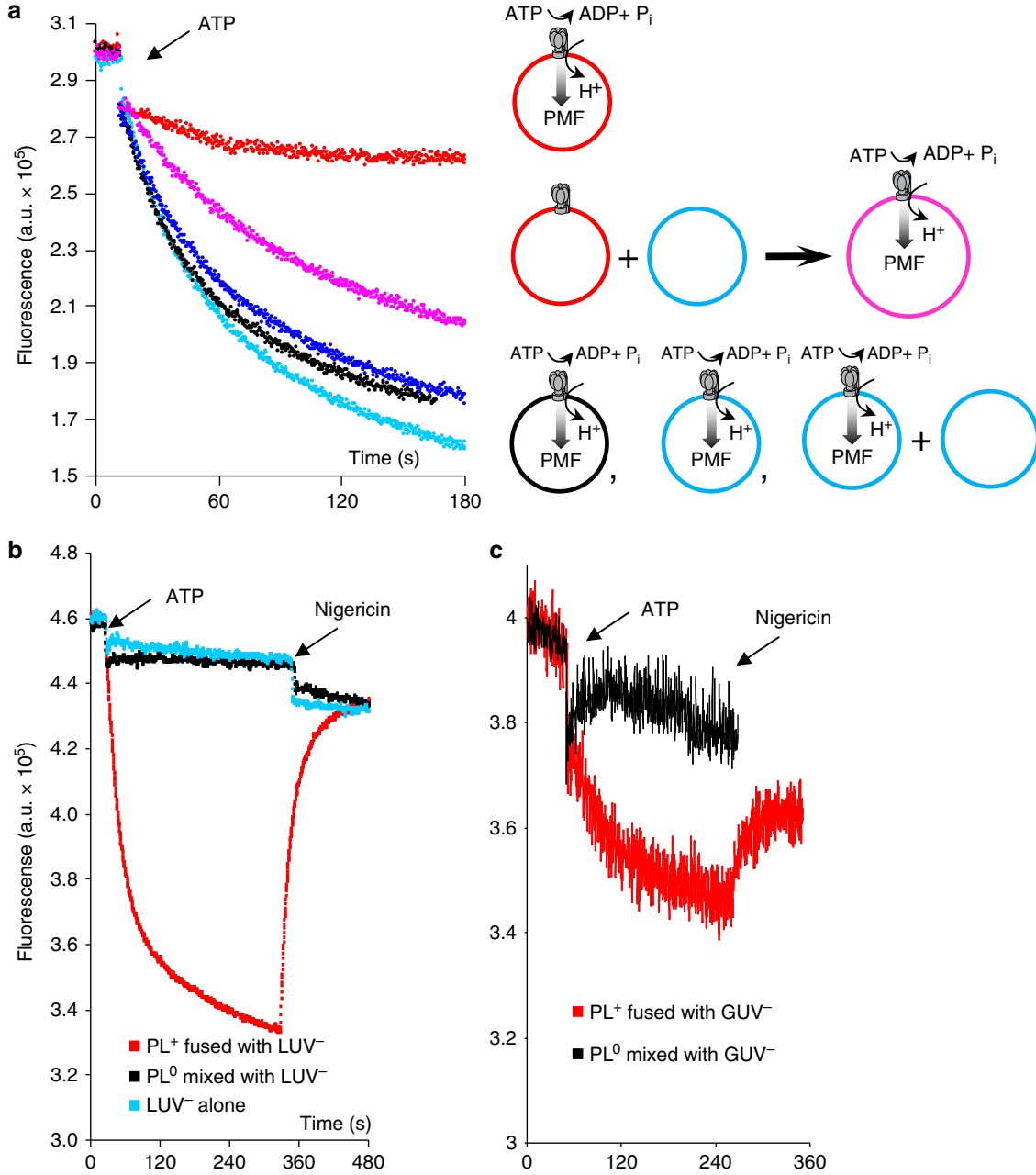

**Figure 2 | Delivery of functional ATP-synthase into target membranes by vesicle fusion.** (**a**) Pink: proton pumping driven by ATP hydrolysis in ATP-synthase, delivered into anionic SUV by fusion with cationic proteoliposomes (PL$^+$) in 10 mM KCl, 1 mM MgCl$_2$, 1 mM MOPS pH 7.4 and assayed by ACMA quenching in the same buffer. Protons are pumped into the fusion product in the presence of 0.2 mM ATP, acidifying the interior which quenches ACMA fluorescence. Red: PL$^+$ alone show less ACMA quenching without the addition of anionic SUV. Black, blue: neutral or anionic proteoliposomes show more ACMA quenching than PL$^+$. Dark blue: addition of anionic SUV to PL$^-$ has no effect on ACMA quenching. Schematic representation of this experiment is shown in the right panel. (**b,c**) Red: proton pumping driven by ATP hydrolysis in ATP-synthase, delivered into anionic LUV (**b**) or GUV (**c**) by fusion with PL$^+$, assayed by ACMA quenching as described for **a**. Addition of uncoupler nigericin dissipated the proton gradient. Unfused PL$^+$ were removed by low *g* centrifugation. Black: as above, but with PL$^+$ replaced by neutral PL$^0$, which do not fuse with LUV and GUV, and thus ATP-synthase is removed by centrifugation leading to negligible ACMA quenching. Blue: as above, but with no proteoliposomes.

ACMA quenching by ATP hydrolysis in the post-fusion product formed by mixing PL$^+$ with SUV$^-$ for 10 min in a low ionic strength buffer (10 mM KCl, 1 mM MgCl$_2$, 5 mM MOPS, pH 7.4). ACMA quenching in this mixture was much better than in the PL$^+$ alone (Fig. 2a, red), but weaker than in PL$^0$ (black) or PL$^-$ (blue). The increase in ACMA quenching upon addition of PL$^+$ to SUV$^-$ is a clear indicator of fusion, consistent with the previously observed reduction of quenching in cationic vesicles compared with neutral or anionic vesicles. By contrast, adding PL$^-$ to SUV$^-$ had no effect on ACMA quenching (Fig. 2a, dark blue), which indicates that the effect is due to the altered lipid composition of the post-fusion membrane.

Figure 2a demonstrates the principle that we can deliver functional ATP-synthase from PL$^+$ into a target anionic SUV by vesicle fusion. However, the post-fusion product is not very different in size from the original proteoliposomes. Figure 2b,c, red, show ACMA quenching following delivery of functional ATP-synthase from PL$^+$ into much larger 800 nm LUV$^-$

(Fig. 2b), or several micron diameter $GUV^-$ (Fig. 2c). We fused $PL^+$ with empty anionic vesicles as described above and then separated large post-fusion membranes from non-reacted $PL^+$ by pelleting membranes at low g-force values (15,000$g$, 15 min) three times, resuspending each time in fresh buffer. The absence of ACMA quenching when the experiment was repeated with $PL^0$ replacing $PL^+$ (Fig. 2b,c, black), similar to the case with no PL (Fig. 2b, blue), demonstrates that small vesicles were effectively removed by this procedure, and therefore that ACMA quenching is probably occurring in the LUV or GUV post-fusion products. The relatively weak and noisy signal in the case of GUV is explained by the large size and much smaller quantity of GUV compared with SUV and LUV.

**Vesicle fusion-based building of electron transport chain**. We describe modular assembly of complex systems capable of ATP synthesis by charge-based fusion of simple components. Two systems were demonstrated. Figure 3a, top, shows a complementary binary system where we fused $PL^+$ and $PL^-$ containing separately ATP-synthase and $bo_3$-oxidase. Figure 3a, bottom, shows a ternary system, where we fused empty $LUV^-$ or $GUV^-$ with a mixture of separately prepared $PL^+$ containing ATP-synthase or $bo_3$-oxidase.

In the assembled electron transport chain energization of the membrane is triggered by addition of reduced dithiothreitol ($DTT_{red}$), which reduces Coenzyme $Q_1$ (2,3-dimethoxy-5-methyl-6-(3-methyl-2-butenyl)-1,4-benzoquinone) to make it available to $bo_3$-oxidase. Oxidation of reduced $Q_1$ ($Q_1H$) by $bo_3$-oxidase pumps protons into each post-fusion vesicle. This builds up PMF which drives ATP synthesis by ATP-synthase in the same vesicle. ATP synthesis is initiated by adding potassium phosphate ($KP_i$) to energized vesicles 1 min after addition of DTT. The ATP synthesized is in the external medium, and can be detected by luminescence using the luciferin-luciferase system.

The vesicles were fused in 20 mM KCl, 5 mM MOPS (pH 7.4), 1 mM $MgCl_2$ for 7 min. Fusion was stopped by adding KCl and MOPS to final concentrations of 100 and 50 mM, respectively, and the luciferin-luciferase-ADP cocktail prepared as described in Methods, followed by quinone $Q_1$, DTT and $KP_i$. Oxidation of $Q_1$ by the primary proton pump $bo_3$-generates PMF sufficient to maintain stable ATP synthesis by ATP-synthase for 5–7 min until depletion of oxygen, which being the terminal electron acceptor is consumed by $bo_3$-oxidase. Conversion of synthesized ATP into pyrophosphate and AMP by luciferase is followed by oxidation of its substrate luciferin, and generates light, which is registered in a luminometer in a real-time mode. A similar approach was reported previously[17], in which an electron transport chain was assembled using SNARE proteins to drive fusion of SUV containing ATP-synthase with SUV containing various primary pumps.

ATP synthesis by the binary system is shown in Fig. 3b. Post-fusion vesicles formed by complementary SUV demonstrated a continuous high rate of ATP synthesis for $F_1F_o$ $PL^-$ fused with $bo_3$ $PL^+$ (blue trace). For $F_1F_o$ $PL^+$ fused with $bo_3$ $PL^-$ the rate of ATP synthesis was ~2.5 times lower (red). Our estimates of the ATP synthesis rate (up to ~1 μmol ATP $min^{-1} mg^{-1}$ ATP-synthase, $K_{cat}$ ~10/s) are calculated per mg of ATP-synthase used in PL reconstitution. About 50–70% of this is successfully reconstituted into PL[35,47], of which >97% is expected to be correctly oriented[47], and ~20% of that is expected to be delivered by vesicle fusion. Thus the rates per ATP-synthase in the reconstituted electron transport chain may be 7–10 times higher than our conservative estimates, similar to rates previously reported ($K_{cat}$ ~70/s)[48]. Control experiments including the PMF uncoupler nigericin throughout (grey), or a specific inhibitor

DCCD of $F_1F_o$ ATP-synthase (pink), or mixing in 200 mM KCl (green), or mixing non-complementary $F_1F_o$ $PL^-$ and $bo_3$ $PL^0$ (black), showed no ATP synthesis. These data convincingly prove full fusion of complementary charged vesicles, because ATP synthesis is possible only when both proteins are present in the same energized bilayer.

Similar results were observed for the ternary system (Fig. 3d,e). The ATP synthesis rate in post-fusion LUV (0.3 μmol ATP/min $mg^{-1}$ ATP-synthase) was similar to that in SUV, while the rate in post-fusion GUV was 10 times slower (0.03 μmol ATP/min $mg^{-1}$ ATP-synthase). This is not surprising given the much larger diameter of the postfusion vesicles, which may not be energized by $bo_3$-oxidase to the same extent as SUV and LUV, possibly due to lower density of $bo_3$-oxidase in the membrane and/or smaller surface-to-volume ratio.

**Discussion**

Our data demonstrate that complementary charged lipids provide easy and fast one-step delivery of functional large membrane protein complexes into target bilayers of various sizes. Our approach requires minimal preparation or specialist reagents and allows easy modification of such fragile and sensitive lipid bilayer systems as giant unilamellar vesicles. We believe that our method should be applicable to many other transmembrane proteins, to the delivery of aqueous contents to the interior of a GUV, or where it is desired to alter the lipid composition of the membrane. It may be particularly useful in synthetic biology or biotechnology applications, as it allows multi-component systems of arbitrary complexity to be assembled in a modular fashion from simple components. Any combination of $PL^+$ can be added to the fusion mixture, each delivering its own particular membrane protein complex, lipids and contents to the target membrane.

A similar recent report[17] used SNARE protein-mediated vesicle fusion to deliver membrane proteins into accepting bilayers. Our method has two advantages over SNARE-driven fusion. First, it is much faster (3–10 min versus 40–60 min to finish the fusion reaction). Second, it allows delivery of proteins into vesicles of any size (0.1–10 μm tested), while the requirement of complementary proteins in the accepting bilayers for SNARE-driven fusion currently limits this method to vesicles of small size[17]. A small disadvantage of our method is that lipid vesicles need to be mixed in moderately low ionic strength media (up to 40 mM monovalent salt) for a few minutes for fusion to proceed, but this should not be critical since the ionic strength can be adjusted after fusion is finished. In combination with SNARE proteins, including possible use as a method to deliver SNARE proteins into target membranes of large vesicles, our method may become a universal modular tool, substantially extending the number of potential applications.

We have demonstrated here that the method is suitable for the transmembrane proteins $F_1F_o$ ATP-synthase and $bo_3$-oxidase, which are fully functional in the neutral or anionic post-fusion membrane. The cationic $PL^+$ membrane reduces the rate of proton pumping and introduces a dependence upon ionic strength (Supplementary Fig. 8b). To the best of our knowledge this phenomenon has not been previously described and we speculate as to its implications.

Recent cryo-electron microscopy[33] and structural[49] studies of ATP-synthase demonstrated that the key transmembrane helices of subunit *a* (the key subunit of the membrane sector of $F_o$, whose role is to couple proton translocation across the membrane to rotation of the enzyme's rotor by forming a dynamic interface between the moving parts inside the membrane sector of the protein) lie in the plane of the membrane, an unusual departure from the typical arrangement of transmembrane helices perpendicular to the membrane. This may substantially thin the

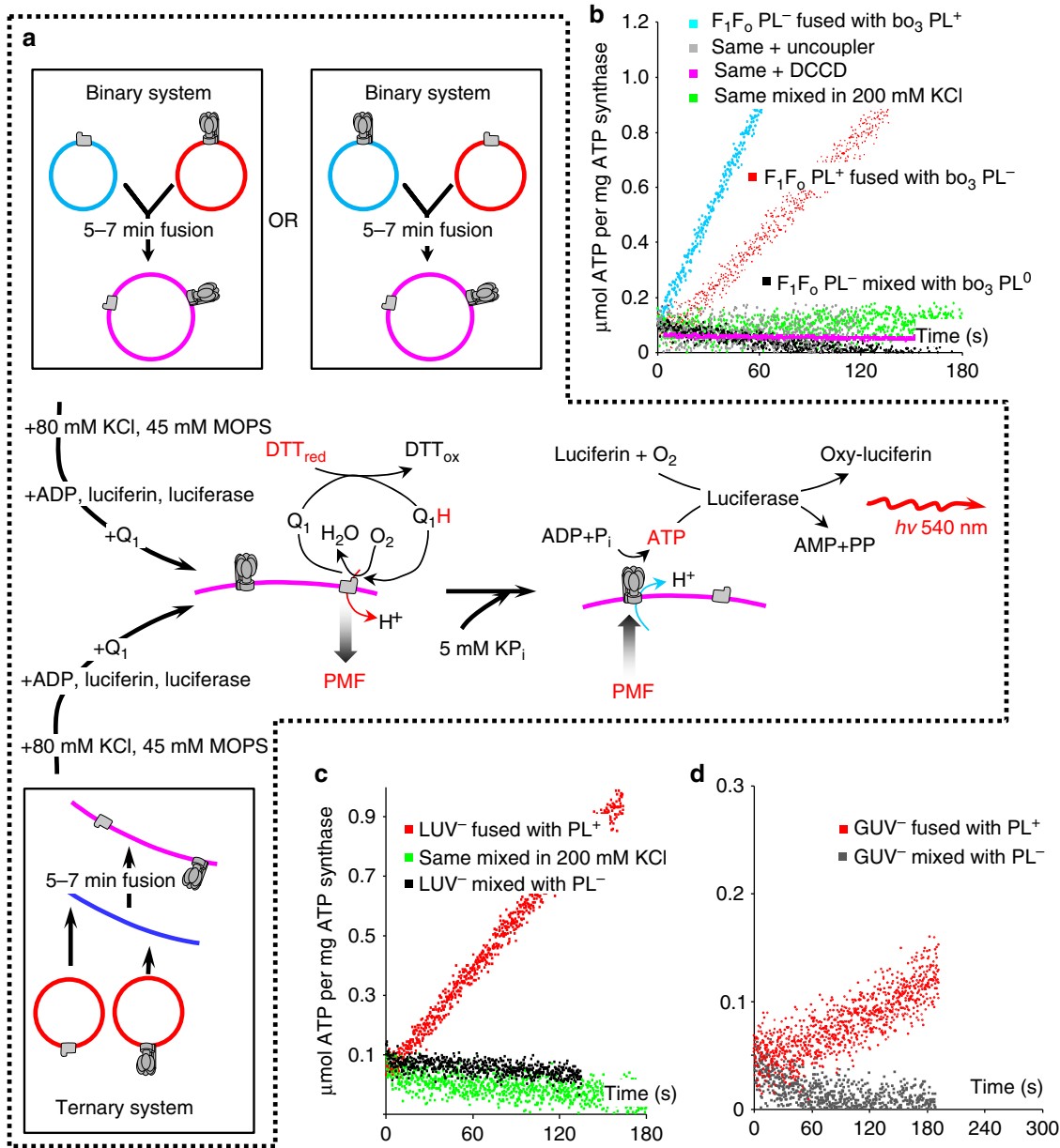

**Figure 3 | Modular assembly of a functional electron transport chain by delivering ATP-synthase and bo₃-oxidase into target membranes.**
(**a**) Schematic representation of the experiment. (top) Binary system (two components): cationic (PL⁺) and anionic proteoliposomes (PL⁻) fused together. (bottom) Ternary system (three components): anionic LUV or GUV fused with two types of PL⁺ containing either ATP-synthase or bo₃-oxidase. The vesicles were fused in 20 mM KCl, 5 mM MOPS pH 7.4, 1 mM MgCl₂ for 5–7 min followed by addition of KCl and MOPS to final concentrations of 100 and 50 mM, respectively, the luciferin-luciferase-ADP cocktail prepared as described in Methods, and oxidized Coenzyme Q₁. In the assembled electron transport chain energization of the membrane is triggered by addition of dithiothreitol (DTT_red), which reduces Q₁ (Q₁H) to make it available to bo₃-oxidase. Oxidation of reduced Q₁ by bo₃-oxidase pumps protons into each post-fusion vesicle. This builds up a PMF which drives ATP synthesis by ATP-synthase in the same vesicle. ATP synthesis is initiated by adding potassium phosphate (KP_i) to energized vesicles 1 min after addition of DTT. The ATP synthesized is detected by the luciferin-luciferase system, where conversion of synthesized ATP into pyrophosphate and AMP by luciferase is followed by light emission registered in a luminometer. (**b**) ATP synthesis by binary system. Blue trace: ATP synthesis by $F_1F_o$ PL⁻ fused with bo₃ PL⁺; grey: same as blue but with nigericin; pink: same as blue but with DCCD-treated $F_1F_o$ PL⁻; green: same as blue but in 200 mM KCl; black: same as blue, but with bo₃ PL⁰;. Red trace: ATP synthesis by $F_1F_o$ PL⁺ fused with bo₃ PL⁻. (**c,d**) ATP synthesis by ternary system. Red trace: ATP synthesis by $F_1F_o$ PL⁺ and bo₃ PL⁺ fused with LUV⁻ or GUV; green: same as red but in 200 mM KCl. Black: PL⁻ were used instead of PL⁺ in the fusion reaction.

hydrophobic barrier of the membrane, making the dynamic interface between the rotor and the stator subunits sensitive to the surface charge, and thus to the poorly screened positive charge of cationic lipids in low ionic strength media. Screening with increasing ionic strength of the medium (Supplementary Fig. 8), or neutralizing by fusion with anionic lipids (Fig. 2a) would then

relieve the destabilizing effect of the cationic lipid charge. In addition it has been demonstrated that the anionic cardiolipin is tightly associated with $F_o$ (ref. 50) and essential for functionality[51]. Excess positive charge in vicinity of the bound cardiolipin may affect enzyme functionality by disturbing the local charge distribution.

## Methods

All the experiments presented here were done with soybean PC mixed with synthetic charged lipids and repeated 2–4 times, with different protein isolations. Results of typical experiments are shown.

**Chemicals.** 1,2-dimyristoleoyl-sn-glycero-3-ethylphosphocholine (E-PC), 1,2-di-(9Z-octadecenoyl)-3-trimethylammonium-propane (DOTAP), 1-palmitoyl-2-oleoyl-sn-glycero-3-phosphate (POPA), and 1,2-dioleoyl-sn-glycero-3-phosphocholine (DOPC) were from Avanti. Lissamine Rhodamine B 1,2-dihexadecanoyl-sn-dlycero-3-phosphoethanolamine (Rho-B), N-(7-nitrobenz-2-oxa-1,3-diazol-4-yl)-1,2-dihexadecanoyl-sn-glycero-3-phospho-ethanolamine (NBD), cholesteryl-4,4-difluoro-5,7-dimethyl-4-dora-3a,4a-diaza-s-dndacene-3-dodecanoate (BODIPY-FL) were from LifeTechnologies. Poly(L-lysine)-g-poly(ethylene glycol) (PLL-PEG) and nitrilotriacetic acid functionalized poly(L-lysine)-g-poly(ethylene glycol) (PLL-PEG-NTA) were from SuSoS. Paraffin oil, 2,3-dimethoxy-5-methyl-6-(3-methyl-2-butenyl)-1,4-benzoquinone (Coenzyme $Q_1$), N,N-dimethyldodecylamine N-oxide (LDAO), dithiothreitol (DTT), N,N′-dicyclohexylcarbodiimide (DCCD), luciferin, p-aminobenzamidine, ATP and ADP were from Sigma-Aldrich. Luciferase was from Roche Diagnostics. All other chemicals were of the highest purity grade available. Glass slides and coverslips were from Menzel-Glaser.

All the procedures and assays described below were done at room temperature (∼22 °C) unless indicated.

**Formation of SUV and LUV.** Lipids were stored in chloroform at −20 °C. BODIPY-FL, NBD and Rho-B were purchased as a solid, suspended in chloroform at 1 mg/ml and added as 0.5% weight fraction (BODIPY-FL) or 2% (NBD or Rho-B) to lipids when indicated. Generally, 10 mg of lipid dissolved in chloroform was added to a glass vial and evaporated under a nitrogen stream followed by vacuum for 10 min. 1 ml of buffer A (100 mM KCl, 1 mM $MgCl_2$, 50 mM MOPS, pH 7.4) was then added to the vial for 30 min to hydrate the lipid followed by thorough vortexing to resuspend the mixture. Vesicle formation was performed by extrusion with a 100, 200 and 800 nm pore size (Whatman Nucleopore Track Etch Membrane); with the sample passed between a filter 21 times using an extrusion system with two 1 ml syringes (Avanti extruder).

In this study, we used lipid mixtures prepared either from pure synthetic lipids or natural lipid extracts; mixtures of both types demonstrated similar results.

**GUV formation.** GUV were formed by an inverted emulsion method[52]. Anionic lipid mixture (20 μl of soybean PC chloroform stock at 100 mg ml$^{-1}$ and 20 μl of POPA chloroform stock at 25 mg ml$^{-1}$) was first placed in 1 ml of paraffin oil and held under constant mixing and heating at 80 °C for 30 min to evaporate chloroform. 200 μl of this mixture was placed on top of 0.5 ml of buffer B (20 mM KCl, 0.1 mM $MgCl_2$, 10 mM MOPS pH 7.4) in an Eppendorf tube and incubated for 1 h to form the oil–water interface. Concurrently, 100 μl of the oil–lipid mixture was mixed with 0.5 μl of aqueous solution containing buffer B with 15% Ficoll-400 (w/v) in a second Eppendorf tube. To form a water-in-oil emulsion this mixture was sonicated for 30 s in an ultrasonic water bath and then vigorously mixed by vortexing for 45 min. The resultant emulsion was placed on top of the oil–water interface and immediately centrifuged in a table-top centrifuge for 2 min at 10,000g. The resulting pellet of GUV was resuspended in 50 μl of fresh buffer B after oil was carefully removed from the tube.

**Calibration of fluorescence signals.** The fluorescence assays for vesicle fusion, liquid content leakage, intervesicular lipid mixing and inner lipid monolayer mixing (below) were calibrated as follows. After fusion was finished, detergent (Triton X-100) was added to a final concentration of 0.05% to release all fluorophores in vesicles. For vesicle fusion only, 7.4 mM EDTA was added with detergent to free all calcein from cobalt (Supplementary Fig. 1). Calibrated fluorescence signals were defined as % of the maximum following detergent addition.

**Demonstration of vesicle fusion by cobalt-calcein.** 200 nm cationic (SUV$^+$) or neutral (SUV$^0$) liposomes were formed in 1 mM calcein, 1 mM $CoCl_2$, 90 mM NaCl, 10 mM MOPS, pH 7.4 at 5 mg ml$^{-1}$ of lipid. 200 nm anionic liposomes (SUV$^-$) were formed in 10 mM EDTA, 80 mM NaCl, 10 mM MOPS pH 7.4 at 5 mg ml$^{-1}$ of lipid. Extruded SUV were pelleted three times at 1,000,000g for 20 min, with resuspension in 1 ml of buffer C (100 mM KCl, 10 mM MOPS pH 7.4) twice, and 0.6 ml buffer C after the last round of pelleting. This removes most external cobalt-calcein and EDTA. To remove the remaining external calcein and EDTA, SUV were passed through a disposable gravity-flow column loaded with Superfine Sephadex G-50 equilibrated with buffer C. This was enough to remove all external cobalt-calcein from SUV$^0$ as judged by a very pale orange colour of the SUV pellet. In contrast SUV$^+$ apparently still had some cobalt-calcein bound to their surface since the pellet colour was bright orange. To further minimize release of the surface bound cobalt-calcein and EDTA, before fusion SUV (typically 5 μl per reaction) of all types were incubated in 1 ml of buffer D (1 mM MOPS, pH 7.4) supplemented with 0.2 mM $CoCl_2$ and the desired concentration of KCl for at least 1 h. The actual extent of fusion may be underestimated due to release of remaining

calcein associated with the lipid surface rather than free calcein inside vesicles, upon addition of detergent.

The reaction was started by mixing 1 ml of SUV$^+$ or SUV$^0$ with 1 ml of SUV$^-$ in a 2 ml fluorimeter cuvette, and fluorescence of cobalt-free calcein measured using 480 nm excitation and 510 nm emission (Fig. 1a,b).

Liquid content leakage was monitored as for vesicle fusion except for the following differences. SUV$^-$ contained 100 mM calcein and 10 mM MOPS, pH 7.4; SUV$^+$ and SUV$^0$ contained no probes, as described[53]. The reaction was started by mixing 2 μl of SUV$^-$, 10 μl of SUV$^+$ or SUV$^0$ and 2 ml buffer D plus indicated KCl concentrations. Calcein fluorescence is self-quenched at high concentrations, and increases upon release from vesicles.

Intervesicular lipid mixing was monitored as for vesicle fusion except for the following differences. SUV$^-$ contained FRET (Forster Resonance Energy Transfer) fluorescence donor NBD and acceptor Rho-B at 2% weight fraction. SUV$^+$ and SUV$^0$ contained no probes. SUV were centrifuged for 5 min at 6,000g to remove clumps and washed once only in buffer C, with no passage through a column. The reaction was started by mixing 5 μl of SUV$^-$, 20 μl of SUV$^+$ or SUV$^0$ and 2 ml buffer D plus indicated KCl concentrations. Donor fluorescence increases[53] upon dilution of acceptor, either by vesicle fusion or detergent addition, were measured using 465 nm excitation and 530 nm emission.

Inner lipid monolayer mixing was monitored as for intervesicular lipid mixing except for the following differences. Before mixing with SUV$^+$, donor fluorescence in the outer monolayer of SUV$^-$ was quenched[54] by mixing 100 μl of SUV$^-$ with 560 μl of buffer C plus 10 mM sodium dithionite ($Na_2S_2O_4$, prepared as 1 M stock in 50 mM TRIS, pH 7.4), which reduces NBD to a non-fluorescent analogue. ∼50% of NBD fluorescence was quenched within 1–2 min with no further quenching over ∼20 min, indicating nearly complete quenching in the outer monolayer only. Immediately after quenching, to remove free $Na_2S_2O_4$, SUV$^-$ were passed through a Sephadex G-50 column and pelleted as described above for vesicle fusion, and resuspended in 100 μl of buffer C.

**Protein purification and reconstitution into SUV.** Histidine-tagged $F_1F_o$ and bo$_3$ oxidase were expressed from pFV2 (ref. 35) and pJRHisA[36] plasmids in DK-8 and C43(DE3) strains of *E. coli*, respectively. Proteins were purified using a slightly modified procedure[35] as follows. Membranes obtained by French pressing were solubilized in extraction buffer containing 50 mM Tris/HCl, pH 7.5, 100 mM KCl, 40 mM ε-aminocaproic acid, 15 mM p-aminobenzamidine, 5 mM $MgCl_2$, 0.8% phosphatidylcholine, 1.5% octyl glucoside, 0.5% sodium deoxycholate, 0.5% sodium cholate, 2.5% glycerol, and 30 mM imidazole at 4 °C for 90 min followed by a 30 min centrifugation at 1,000,000g to separate non-solubilized material. The supernatant was loaded onto a Ni-NTA gravity-flow column equilibrated with extraction buffer, and eluted with extraction buffer containing 180 mM imidazole.

An amount of 100 μg of purified protein was used for reconstitution into extrusion-formed 100 nm SUV in buffer A. For that 100 μl of protein at 1 mg ml$^{-1}$ in extraction buffer was mixed with 60 μl 10% sodium cholate, 300 μl SUV and 200 μl buffer A on a rocking platform at 4 °C for 15 min and passed through 3 ml of Sephadex G-50 resin packed into a disposable plastic gravity flow column equilibrated with buffer A at room temperature. The turbid fraction was pooled, and proteoliposomes were pelleted once at 450,000g and resuspended in 1 ml of buffer A.

**Demonstration of fusion in a microscope tunnel slide.** A tunnel slide was prepared by mounting a cover slip on a glass slide with 100 μm double-sided sticky tape. Glass surface was modified by loading the slide with a mixture of PLL-PEG (1 mg ml$^{-1}$) with PLL-PEG-NTA (0.1 mg ml$^{-1}$) in a buffer containing 15 mM HEPES, pH 5.5 for 10 min followed by wash with the same buffer containing 10 mM NiCl$_2$, and rinsed with buffer A. BODIPY-FL cholesterol $F_1F_o$ PL$^+$ were diluted 250 times in buffer A, loaded into the slide for 5 min followed by wash with buffer B. GUV$^-$ were loaded slowly into the slide, left for 10 min to fuse, and then free GUV were gently washed away. Postfusion vesicles were inspected using ×100 NA 1.45 oil immersion objective on an inverted Nikon Eclipse TE2000U microscope equipped with a mercury arc lamp, a commercial fluorescence filter set and a charge-coupled device digital camera (Thorlabs 340M-GE).

**ACMA quenching.** A total of 40 μl of PL were added to 2 ml of buffer A or to 1 mM MOPS, 1 mM $MgCl_2$, pH 7.4, and the indicated concentration of KCl, in the presence of 0.5 μM ACMA. The reaction was followed in a fluorimeter at 430 nm excitation and 515 nm. Once a stable signal was observed, the reaction was initiated by addition of 0.2 mM ATP to ATP-synthase PL, or 40 μM Coenzyme $Q_1$ followed by 2 mM DTT to bo$_3$ oxidase PL, and finally stopped by 2 μM uncoupler nigericin.

**Assembly of electron transport chain and ATP synthesis.** The binary (3 μl of bo$_3$-oxidase PL and 5 μl of $F_1F_o$ PL formed as described above) or ternary (abovementioned volumes of proteoliposomes and 3 μl of LUV or GUV) system was added to 800 μl of buffer E (16 mM KCl, 1 mM $MgCl_2$, 3 mM MOPS, pH 7.4) and fused for 5–7 min, and then fusion was stopped by raising KCl and MOPS to 100 and 50 mM, respectively, with 18 μl of 4 M KCl and 39 μ 1 M MOPS, pH 8. This mixture was added to 200 μl luciferin-luciferase-ADP cocktail (400 μM ADP, 50 μM luciferin, 2.5 μg luciferase in 100 mM KCl, 1 mM $MgCl_2$, 50 mM MOPS pH

7.4). MgCl$_2$ was added to this mixture to a final concentration of 3 mM, followed by 40 μM Coenzyme Q$_1$. Energization of post-fusion membranes was triggered by addition of 2 mM DTT, which reduces Q$_1$ to make it available to bo$_3$-oxidase. ATP synthesis was initiated by adding 5 mM potassium phosphate (KP$_i$, pH 7.4) to energized vesicles 1 min after addition of DTT. This gave a stable ATP synthesis reaction, which would last for 5–7 min until depletion of oxygen consumed by bo$_3$-oxidase and luciferase. The ATP synthesized in the course of the reaction is detected by the luciferin-luciferase system, where conversion of synthesized ATP into pyrophosphate and AMP by luciferase is followed by light emission registered in a luminometer (Sirius-L single tube luminometer, Titertek). After the reaction is finished an ATP reference standard (0.5 nmol ATP) was added twice. To obtain the actual amount of ATP produced in the reaction the signal from ATP synthesis reaction was divided by the ATP reference standard signal. This value was adjusted to the total amount of F$_1$F$_o$ protein used in forming PL, and finally expressed as the rate of ATP synthesis in μmol ATP/mg F$_1$F$_o$ /min.

**Estimation of ATP synthase orientation in proteoliposomes.** A total of 30 μl PL$^0$ were mixed with 1.5 μl of 10% sodium cholate in buffer A for 30 s, and then added to 2 ml of a medium containing ATP regenerating system (100 mM KCl, 50 mM MOPS, pH 7.4, 2.5 mM MgCl$_2$, 1 mM ATP, 2 μM nigericin, 2 mM phosphoenolpyruvate, 0.2 mM NADH, 5 units per ml of pyruvate kinase and lactate dehydrogenase). ATP hydrolase activity followed at 340 nm showed no activation upon addition of cholate (Supplementary Fig. 9a).

**DCCD inhibition of ATP-synthase.** DCCD, a covalent modifying agent of Asp61, the key amino acid residue responsible for proton translocation of subunit c of ATP-synthase[55], specifically inhibits ATP synthesis at 50 μM. 40 μl F$_1$F$_o$ PL$^0$ and bo$_3$ PL$^0$ were incubated with 50 μM DCCD in 2 ml of buffer A for 60–90 min and tested for substrate-driven ACMA quenching before fusion as described above. As expected, DCCD abolished ATP synthesis in post-fusion electron transport complexes (Fig. 3b, pink) and ATP hydrolysis driven ACMA quenching by ATP-synthase (Supplementary Fig. 9b) but did not affect ACMA quenching by bo$_3$-oxidase.

**Data availability.** The authors declare that all data supporting the findings of this study are available within the article and its Supplementary Figures File, or from the corresponding authors upon request.

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

## Acknowledgements

We are thankful to Drs Robert Gennis from University of Illinois and Christoph von Ballmoos from University of Bern for providing a plasmid and a strain to express $bo_3$-oxidase and numerous advices on how to isolate and handle the protein. The project was supported by BBSRC grant BB/L01985X/1 to R.B. and R.I.

## Author contributions

R.R.I. and R.M.B. conceived the research project. A.N.R. and R.R.I. conducted experiments and analysed the data. All the authors contributed to discussion of the results for the manuscript. R.R.I., A.N.R. and R.M.B. wrote the manuscript.

## Additional information

**Competing financial interests:** The authors declare no competing financial interests.

