## [Peer review file · Nature Communications]

Reviewer #1 (Remarks to the Author)

The manuscript from Ishmukhametov et al is very interesting in that it demonstrates the possibility to use fusogenic proteoliposomes containing charged lipids and membrane proteins, separately reconstituted, for a co-reconstitution. These proteoliposomes fuse thanks to oppositely charged bilayers without the need of detergent or of fusion-promoting proteins. The data are already rather convincing.

However, there is a peculiar assumption, missing simple controls and missing information.

Peculiar assumption: in all the drawings and in a sentence on p7 line 156, it is assumed that the ATP-synthase is fully oriented (so as to pump into proteoliposomes) ; the same assumption is made for the drawing of bo3-oxidase (drawing in fig 3). Nothing is showing that it is the case. Of course the addition of substrates is such that the activity can only be measured with this single orientation, but the experience is that a 100% orientation is very difficult to reach after membrane protein reconstitution and cannot be simply assumed and it is important to discuss this possibility. The assumption has an impact on the specific activity measurements. In that respect , the specific activity measured for the ATP synthesis is on the low side if I assume that the t° of measurement was 37°C (see below) I calculate catalytic turnover of the order of 10/sec, but could be doubled if the orientation is 50/50 or if some proteoliposomes are leaky (ATP consumption?).

Missing simple controls: ATP synthesis experiments should be tested in the presence of the inhibitors of FOF1 (DCCD) and even more importantly that the system is insensitive to the inhibitors of kinases (AP5A).

Missing information: in almost all the procedures the t° is not mentioned , of course an essential information for enzymatic activity in particular.

Minor remarks

The sentence p4 line 94: " Vesicles compositions...other" is" not clear.

P5 line 105 , it is not correct to write in that sentence that Fig 1B shows fluorescence intensity " since in the fig the data is presented only in terms of " % fusion" .

Some fig could be improved:

- in fig 1 b , 3b and 3c : use larger symbols to visualize the color codes (use the sizes used in fig 3d or S-fig5).

- in fig 1 e , it would be nice to indicate GUV- on top and PL+ on the bottom vesicle.

Legend to S-Fig4: MOPS

Reviewer #2 (Remarks to the Author)

The authors describe a method to induce fusion of proteoliposomes using electrically charged lipids to facilitate vesicle-vesicle interaction. The method is intended to be used in the generation of cell-like artificial structures. The work is not particularly original, since not only charged lipids but also other methods (polyethylene glycol, phospholipase C, several proteins) have been used in the past to induce very fast (a few seconds) vesicle fusion (eg. Sengupta, Biophys J. 2014 Sep 16;107(6):1318, or Lete, Biophys J. 2015 Feb 17;108(4):863).

One serious drawback with this work is that, to demonstrate real fusion of lipid vesicles, all four of the following must be experimentally and individually tested: vesicle aggregation, intervesicular lipid mixing, intervesicular mixing of aqueous contents, and lack of aqueous contents leakage.

Only the third of these assays has been performed in this work. Thus the results cannot be properly interpreted, let alone accepted at their face value, until these uncertainties have been solved. In particular this would solve the hemifusion uncertainty.

Minor points:

-"Complementary" is misspelled throughout the paper. Page 4, line 86, "quinoles/quinols". "15 000 G/15 000 x g" in various parts. sec for s, in various parts and figures.

Reviewer #1.

(1) In all the drawings and in a sentence on p7 line 156, it is assumed that the ATP-synthase is fully oriented (so as to pump into proteoliposomes); the same assumption is made for the drawing of bo3-oxidase (drawing in fig 3). Nothing is showing that it is the case. Of course the addition of substrates is such that the activity can only be measured with this single orientation, but the experience is that a 100% orientation is very difficult to reach after membrane protein reconstitution and cannot be simply assumed and it is important to discuss this possibility... The assumption has an impact on the specific activity measurements. In that respect , the specific activity measured for the ATP synthesis is on the low side if I assume that the t{degree sign} of measurement was 37{degree sign}C (see below) I calculate catalytic turnover of the order of 10/sec, but could be doubled if the orientation is 50/50 or if some proteoliposomes are leaky (ATP consumption?).

We agree. Our new data demonstrate that full fusion happens in 20% of all vesicles, that only ~5% of vesicles are leaky. From previous work referenced in the text (ref 47) it is known that 50-70% protein goes into proteoliposomes. Therefore the actual ATP synthesis rate may be 7-10 times higher than our conservative estimates in Figure 3, i.e. 70-100/sec in agreement with previous estimates referenced in the text (ref 48). Orientation in the membrane, estimates of specific activity and their comparison to literature are now addressed in the main text, page 10, paragraph 3.

(2) ATP synthesis experiments should be tested in the presence of the inhibitors of FOF1 (DCCD)...

We have done the appropriate control experiments, now including DCCD inhibition as well. As expected, DCCD specifically blocked F_1F_0 : there is no ATP-driven proton pumping in ATP synthase containing proteoliposomes, and no ATP production by postfusion vesicles formed by F_1F_0 proteoliposomes treated with DCCD prior to fusion (pink trace in Fig. 3b). By contrast, DCCD-treated bo_3 proteoliposomes showed no inhibition of Q_1 -oxidation driven proton pumping.

(2a) ...and even more importantly that the system is insensitive to the inhibitors of kinases (AP5A).

The Reviewer is concerned that activity of the enzyme adenylate kinase ($2 \text{ ADP} \rightarrow \text{ATP} + \text{AMP}$), present as an impurity in proteoliposomes, may interfere with the luciferase assay and contribute a false positive signal. However, if this were the true we would observe ATP

production without phosphate or energization of the post-fusion vesicles, or in the presence of an uncoupler. But this is not the case: ATP synthesis by postfusion vesicles was initiated by addition of phosphate, and as Figure 3,B demonstrates there was no ATP synthesis in the presence of uncoupler nigericin (grey trace) nor without energization of F_1F_0 in non-fused vesicles (green and black traces).

(3) in almost all the procedures the temperature is not mentioned , of course an essential information for enzymatic activity in particular.

Now it is corrected. We introduced a sentence in Methods section, which states (page 13, paragraph 2): All the procedures and assays described below were done at room temperature (~22° C).

(4) The sentence p4 line 94: "Vesicles compositions...other" is" not clear.

The wording is improved, and now (page 4, paragraph 2) reads: "The lipid compositions were designed to form vesicles that fuse when mixed in the combination 2+3 but no other."

(5) P5 line 105 , it is not correct to write in that sentence that Fig 1B shows fluorescence intensity " since in the fig the data is presented only in terms of " % fusion" .

It reads now (page 5, paragraph 1): "Figure 1b, red shows vesicle fusion vs time following mixing of cationic and anionic 200 nm SUV in a dilute buffer (1 mM MOPS pH 7.4), indicating rapid vesicle fusion. Fusion (%) was calculated from calcein fluorescence intensity, and calibrated by adding detergent to release and thus mix all vesicle contents (Methods, Supplementary Fig. 1)."

(6) - in fig 1 b , 3b and 3c : use larger symbols to visualize the color codes (use the sizes used in fig 3d or S-fig5.

Done; now all the symbols are enlarged for better appearance.

(7) - in fig 1 e , it would be nice to indicate GUV- on top and PL+ on the bottom vesicle.

Done as suggested

(8) Legend to S-Fig4: MOPS

Corrected

Reviewer #2.

(1) ...to demonstrate real fusion of lipid vesicles, all four of the following must be experimentally and individually tested: vesicle aggregation, intervesicular lipid mixing, intervesicular mixing of aqueous contents, and lack of aqueous contents leakage. Only the third of these assays has been performed in this work. Thus the results cannot be properly interpreted, let alone accepted at their face value, until these uncertainties have been solved. In particular this would solve the hemifusion uncertainty.

Vesicle aggregation was already shown in the previous version of the paper (now Figure S4), the reviewer may have missed this. All the other experiments are done now, and in addition we also tested lipid mixing of inner lipid monolayers. All the new data are in good agreement, indicating that we provide convincing evidence of vesicle fusion as based on characterisation of their liquid content transfer, liquid content leakage, and mixing of their lipids, including inner lipid monolayer mixing. New results are shown in the new manuscript, page 5, paragraph 3 to page 6, paragraphs 1-2. We also updated the Methods section to include description of the new experiments.

(2) The work is not particularly original, since not only charged lipids but also other methods (polyethylene glycol, phospholipase C, several proteins) have been used in the past to induce very fast (a few seconds) vesicle fusion (eg. Sengupta, Biophys J. 2014 Sep 16;107(6):1318, or Lete, Biophys J. 2015 Feb 17;108(4):863).

We have referred to the paper by Lete et al (new ref 53) in the new methods sections describing our new measurements of liquid content release and lipid mixing. We believe that the paper by Sengupta et al reports a highly specialized method of fusion and therefore is not directly relevant to our manuscript. We believe that the novelty of our results is sufficiently described in our manuscript.

(3) "Complementary" is misspelled throughout the paper.

Corrected.

(4) Page 4, line 86, "quinoles/quinols". "15 000 G/15 000 x g" in various parts. sec for s, in various parts and figures.

Now all these are corrected throughout the main body text and figure legends.

Reviewer #1 (Remarks to the Author)

The authors have adequately answered to my questions/remarks except that they only partially answered in my very first remark ; below is my original question , their reply and what I consider important and for which they did not reply at all.

(1) In all the drawings and in a sentence on p7 line 156, it is assumed that the ATP-synthase is fully oriented (so as to pump into proteoliposomes); the same assumption is made for the drawing of bo3-oxidase (drawing in fig 3). Nothing is showing that it is the case. Of course the addition of substrates is such that the activity can only be measured with this single orientation, but the experience is that a 100% orientation is very difficult to reach after membrane protein reconstitution and cannot be simply assumed and it is important to discuss this possibility... The assumption has an impact on the specific activity measurements. In that respect , the specific activity measured for the ATP synthesis is on the low side if I assume that the $t\{\text{degree sign}\}$ of measurement was $37\{\text{degree sign}\}C$ (see below) I calculate catalytic turnover of the order of 10/sec, but could be doubled if the orientation is 50/50 or if some proteoliposomes are leaky (ATP consumption?).

We agree. Our new data demonstrate that full fusion happens in 20% of all vesicles, that only ~5% of vesicles are leaky. From previous work referenced in the text (ref 47) it is known that 50-70% protein goes into proteoliposomes. Therefore the actual ATP synthesis rate may be 7-10 times higher than our conservative estimates in Figure 3, i.e. 70-100/sec in agreement with previous estimates referenced in the text (ref 48). Orientation in the membrane, estimates of specific activity and their comparison to literature are now addressed in the main text, page 10, paragraph 3.

Here is the part they did not adequately reply in the above answer :) In all the drawings and in a sentence on p7 line 156, (now page 8 line 192) it is assumed that the ATP-synthase is fully oriented (so as to pump into proteoliposomes); the same assumption is made for the drawing of bo3-oxidase (drawing in fig 3). Nothing is showing that it is the case. Of course the addition of substrates is such that the activity can only be measured with this single orientation, but the experience is that a 100% orientation is very difficult to reach after membrane protein reconstitution and cannot be simply assumed and it is important to discuss this possibility... So the authors should have two options : they could write before the first figure is called (e.g. page 8 line 192) " In the present paper we assumed that all the reconstituted proteins are 100% oriented i.e. , e.g., with the ATP/ADP/Pi binding sites of ATPsynthase at the exterior to the vesicles. This assumption is based on ... (refs to be given)". But the best would be if they actually perform this experiment which could demonstrate it. The experiment is rather simple : measure the ATPase activity with the reconstituted proteoliposomes, then add a non-denaturing detergent which will make the inside of the vesicle accessible to the ATP, and see if the ATPase activity is stable (i.e; meaning a 100% orientation as presented in all their drawings) or if it increases (meaning less than 100%).

Reviewer #2 (Remarks to the Author)

The revised version of the manuscript is acceptable for publication.

Reviewer #1.

The authors have adequately answered to my questions/remarks except that they only partially answered in my very first remark; below is my original question, their reply and what I consider important and for which they did not reply at all.

>(1) In all the drawings and in a sentence on p7 line 156, it is assumed that the ATP-synthase is fully oriented (so as to pump into proteoliposomes); the same assumption is made for the drawing of bo3-oxidase (drawing in fig 3). Nothing is showing that it is the case. Of course the addition of substrates is such that the activity can only be measured with this single orientation, but the experience is that a 100% orientation is very difficult to reach after membrane protein reconstitution and cannot be simply assumed and it is important to discuss this possibility... The assumption has an impact on the specific activity measurements. In that respect, the specific activity measured for the ATP synthesis is on the low side if I assume that the t {degree sign} of measurement was 37{degree sign}C (see below) I calculate catalytic turnover of the order of 10/sec, but could be doubled if the orientation is 50/50 or if some proteoliposomes are leaky (ATP consumption?).

>We agree. Our new data demonstrate that full fusion happens in 20% of all vesicles, that only ~5% of vesicles are leaky. From previous work referenced in the text (ref 47) it is known that 50-70% protein goes into proteoliposomes. Therefore the actual ATP synthesis rate may be 7-10 times higher than our conservative estimates in Figure 3, i.e. 70-100/sec in agreement with previous estimates referenced in the text (ref 48). Orientation in the membrane, estimates of specific activity and their comparison to literature are now addressed in the main text, page 10, paragraph 3.

Here is the part they did not adequately reply in the above answer. In all the drawings and in a sentence on p7 line 156, (now page 8 line 192) it is assumed that the ATP-synthase is fully oriented (so as to pump into proteoliposomes); the same assumption is made for the drawing of bo3-oxidase (drawing in fig 3). Nothing is showing that it is the case. Of course the addition of substrates is such that the activity can only be measured with this single orientation, but the experience is that a 100% orientation is very difficult to reach after membrane protein reconstitution and cannot be simply assumed and it is important to discuss this possibility...

So the authors should have two options : they could write before the first figure is called (e.g. page 8 line 192) " In the present paper we assumed that all the reconstituted proteins are 100% oriented i.e. , e.g., with the ATP/ADP/Pi binding sites of ATPsynthase at the exterior to the vesicles. This assumption is based on ...(refs to be given)". But the best would be if they actually perform this experiment which could demonstrate it. The experiment is rather simple: measure the ATPase activity with the reconstituted proteoliposomes, then add a non-denaturing detergent which will make the inside of the vesicle accessible to the ATP, and see if the ATPase activity is stable (i.e; meaning a 100% orientation as presented in all their drawings) or if it increases (meaning less than 100%).

We have addressed this point experimentally by measuring ATP hydrolase activity of freshly made proteoliposomes, with and without non-denaturing detergent (0.5% sodium cholate)

The method is described in a new paragraph of the Methods section (now page 19, paragraph 3): **'Estimation of ATP synthase orientation in proteoliposomes. ATP hydrolase activity by PL**

was measured essentially as described³⁵ in a medium containing 100 mM KCl, 2 mM ATP, 1 mM MgCl₂, 4 μM FCCP, pH indicator 60 μM phenol red (OD₅₅₇ ~ 2), pH 8, with and without 0.5% sodium cholate. The reaction was started by adding 50 μl of freshly made PL, and calibrated by adding 0.5 μmol HCl. Slight acidification due to cholate was compensated by KOH.'

We did not see any increase in ATP hydrolase activity in presence of the detergent, supporting our assumption that nearly all reconstituted ATP synthase in proteoliposomes has outward facing F₁. Our data confirm a previous similar observation (ref 47) that 97 % of reconstituted ATP-synthase in proteoliposomes faces outwards using a similar reconstitution procedure to ours. We describe this new result on page 8, last paragraph: 'After reconstitution (~95 %) of ATP-synthase is oriented so as to pump protons into proteoliposomes when driven by ATP hydrolysis, *i.e* with F₁ complex facing outwards the vesicles⁴⁷. To confirm this we quantified ATP hydrolase activity of freshly made proteoliposomes with and without vesicle disruption by the non-denaturing detergent 0.5% sodium cholate (Methods). We saw no increase with detergent, indicating the absence of an inward facing fraction of F₁ that would have been released by vesicle disruption (data not shown).